## PROCEEDINGS A

quantum physics

entanglement, entanglement swapping, hybrid entanglement, quantum optics, quantum communications

**Author for correspondence:**
Ryan C. Parker
e-mail: ryanparker1447@gmail.com

†Present address: BT Research, Polaris House, Adastral Park, Ipswich, IP5 3RE, UK.

One contribution to a special feature 'Innovative and emerging Communications Concepts and Technologies' organized by Ben Allen and Anas Al Rawi.

# Photonic hybrid state entanglement swapping using cat state superpositions

Ryan C. Parker[1,†,], Jaewoo Joo[2] and Timothy P. Spiller[1]

[1]York Centre for Quantum Technologies, Department of Physics, University of York, York YO10 5DD, UK
[2]School of Mathematics and Physics, University of Portsmouth, Portsmouth PO1 3QL, UK

RCP, 0000-0002-0440-1034

We propose the use of hybrid entanglement in an entanglement swapping protocol, as means of distributing a Bell state with high fidelity to two parties. The hybrid entanglement used in this work is described as a discrete variable (Fock state) and a continuous variable (cat state superposition) entangled state. We model equal and unequal levels of photonic loss between the two propagating continuous variable modes, before detecting these states via a projective vacuum-one-photon measurement, and the other mode via balanced homodyne detection. We investigate homodyne measurement imperfections, and the associated success probability of the measurement schemes chosen in this protocol. We show that our entanglement swapping scheme is resilient to low levels of photonic losses, as well as low levels of averaged unequal losses between the two propagating modes, and show an improvement in this loss resilience over other hybrid entanglement schemes using coherent state superpositions as the propagating modes. Finally, we conclude that our protocol is suitable for potential quantum networking applications which require two nodes to share entanglement separated over a distance of 5–10 km, when used with a suitable entanglement purification scheme.

# 1. Introduction

The future of large-scale quantum communications will almost certainly involve distribution and manipulation of entangled pairs of photons within a quantum network; such a quantum network is likely to include small clusters of quantum processors (perhaps in a local network of quantum computers) which may require shared entanglement, and could then be connected to other network clusters, potentially via satellite communications [1–3]. However, despite the undeniably useful non-classical properties which entanglement-based quantum systems offer (such as for quantum key distribution [4–7], quantum secret sharing [8–10], quantum repeaters [11–14], quantum computing [15–18] and quantum teleportation [19–22]), entanglement is a highly fragile resource, and breaks down rapidly in the presence of noise and losses [23].

One particularly useful proposal for circumventing the intrinsic fragility of distributing entangled photons around a quantum network is by performing entanglement swapping (ES). In ES, there exist two parties, Alice and Bob, each of whom begin the protocol with a separately entangled pair of photons, $AB$ and $CD$, respectively. They each send half of their entangled pair (i.e. modes $B$ and $D$) to a central location, where these propagating modes are mixed at a $50:50$ beam-splitter, before subsequently being measured, as described in the schematic of figure 1. This mixing and measurement step is key to ES, as it causes projection of the states in modes $B$ and $D$, thus projecting modes $A$ and $C$ into an entangled state.

This distributed entanglement, now shared between modes $A$ and $C$, can then be used for further quantum communications and quantum computational purposes. Moreover, performing ES to share entanglement enhances the secrecy and security of the post-entangled state shared between Alice and Bob; if an adversary, Eve, were to measure modes $B$ and $D$, she gains no useful information on states $A$ and $C$, and in fact by carrying out this measurement Eve has actually *assisted* Alice and Bob in sharing an entangled state [24].

This, in fact, is a form of measurement-device independence [25–27], and is a direct consequence of the monogamy of entanglement: if Alice and Bob share an entangled state of high fidelity, then Eve must be disentangled from this state. As a result of this, Alice and Bob need not trust the source of entanglement. However, Eve, as the source of entanglement, could simply deny service, and there is no way to circumvent this form of attack.

ES, first performed experimentally in 1993 [28], has since been executed with discrete variables (DVs) [29–31], and also with continuous variables (CVs) [32,33]. CV systems typically pose the advantage of high success probability, whereas DVs are often robust against lossy channels; hence, an advantage could be gained from using both CV and DV states in what is referred to as a hybrid entanglement scheme [34], and will be investigated in this work.

The practicality of using CVs is also worth noting; they are compatible with current standard optical telecommunication technologies, and so could be suitable for large-scale communication protocols, and could offer greater resistance to photonic losses compared to DV states [35,36]. Moreover, preparation of DV entanglement often requires heralded approaches (whereas CV entangled states are typcially prepared deterministically), and therefore result in lower success probabilities, however, the fidelity against the desired state to be produced often approaches unity [37]. Both, DVs and CVs, have been extensively researched when used in entanglement swapping protocols together in the form of hybrid entanglement [38,39], and have also been demonstrated experimentally [40].

In this work, we present a new ES protocol based on hybrid entangled cat states. We show that we can produce a Bell state of high fidelity when allowing for low levels of photonic losses in the propagating CV (cat state superposition) modes, as well as allowing differences in the losses between the two propagating modes.

This paper is organized as follows: in §2, we introduce our proposed ES protocol, as well as our model for photonic losses in the propagating modes, and the detection methods used; we extend this model for loss, and include averaged unequal losses between modes $B$ and $D$, in §3; in §4, we show that, following our proposed ES protocol, Alice and Bob can share a Bell state of high fidelity, when allowing for low levels of equal and unequal losses, as well as

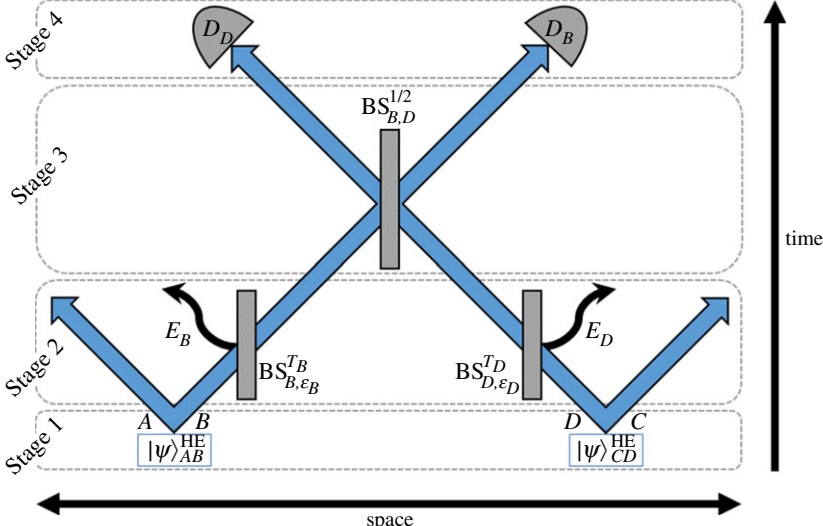

**Figure 1.** Schematic to represent the six-channel system (in which the initial hybrid entangled states are denoted as $|\Psi\rangle_{AB}^{HE}$ and $|\Psi\rangle_{CD}^{HE}$) undergoing entanglement swapping. Modes $A$ and $C$ are the discrete variable post-entangled modes, and $B$ and $D$ are the propagating continuous variable modes. Losses are modelled through leakage of modes $B$ and $D$ into modes $E_B$ and $E_D$, respectively. The lossy modes $B$ and $D$ then meet at a 50 : 50 beam-splitter ($BS_{B,D}^{1/2}$) before subsequently being measured via a projective vacuum measurement and balanced homodyne detection ($D_B$ and $D_D$, respectively). (Online version in colour.)

investigating homodyne measurement imperfections; we discuss the practicality of our protocol for distributing entanglement in a future quantum network in §5; our conclusions are given in §6.

## 2. The entanglement swapping protocol

A space–time schematic describing the process of our proposed ES protocol, from initial hybrid entangled state generation through to detection, is given in figure 1, in which each arrow indicates a channel/mode. Within figure 1, the arrows illustrating modes $A$ and $C$ indicate a movement in space; this is to represent the possibility that these modes may be sent on for further uses in quantum communications protocols or quantum computing, or perhaps even to a potential customer (or indeed separated customers), requiring an entangled pair of qubits.

We now discuss each stage of our proposed protocol, starting with the production of the initial hybrid entangled states, before moving on to discuss how we model photonic losses in the propagating modes, and our subsequent detection methods.

### (a) Hybrid states with superposed cats

In this work, we rely on coherent state superpositions (also referred to as Schrödinger cat states) as the propagating mode of quantum information. A coherent state $|\alpha\rangle$ is the unique eigenstate of the annihilation operator $\hat{a}$, with eigenvalue $\alpha$, thus being described as $\hat{a}|\alpha\rangle = \alpha|\alpha\rangle$, where

$$|\alpha\rangle = e^{-(|\alpha|^2/2)} \sum_{n=0}^{\infty} \frac{\alpha^n}{\sqrt{n!}} |n\rangle, \tag{2.1}$$

in which we have represented the coherent state $|\alpha\rangle$ of amplitude $\alpha$ (where $\alpha$ is complex [41]) as a summation of the discrete variable Fock state $|n\rangle$ [42], which is vital for the analyses required in this work. We define the creation and annihilations operators as $\hat{a} = \hat{x} + i\hat{p}$ and $\hat{a}^\dagger = \hat{x} - i\hat{p}$, respectively, where $\hat{x}$ and $\hat{p}$ are the dimensionless field quadratures which we are

able to measure via homodyne detection (see appendix B), and are described as $\hat{x} = 1/2(\hat{a} + \hat{a}^\dagger)$ and $\hat{p} = (i/2)(\hat{a}^\dagger - \hat{a})$ [43].

Two-component states are coherent state superpositions with opposite phase, and as such we describe the cat states used in this work mathematically as

$$\left.\begin{aligned} |\Psi_\alpha\rangle &= \mathcal{N}_\alpha(|\alpha\rangle + |-\alpha\rangle), \\ |\Psi_{i\alpha}\rangle &= \mathcal{N}_\alpha(|i\alpha\rangle + |-i\alpha\rangle), \end{aligned}\right\} \tag{2.2}$$

where, $\mathcal{N}_\alpha$ is the normalization factor, given as $N_\alpha = 1/\sqrt{2 + 2\,e^{-2\alpha^2}}$. We importantly note that, unless stated otherwise, $\alpha$ is real throughout this work.

We highlight here that freely propagating optical mode cat states have been producible experimentally since their realization in 1986 by Yurke & Stoler [44], via use of the optical Kerr effect, applying a unitary evolution of a single coherent state. More recently however, there have been a multitude of experimental procedures undertaken to produce optical cat states. One such method is via photon subtraction of an optical squeezed state [45–50]; this method is usually suitable for producing low-amplitude cat states, which in fact is more applicable to this work, as we show in our results that cat states of amplitude $1.0 < |\alpha| < 2.5$ yield the best fidelity for our proposed protocol. A further proposed method is given in [51], in which squeezed cat states are shown to be generated from Fock states, via use of a single homodyne detector. For a highly comprehensive review on generating cat states, refer to [52].

Within this ES protocol, we propose the use of so-called hybrid entangled states. These quantum states are described as having entanglement shared between DV and CV degrees of freedom. Extensive experimental research has been carried out preparing various hybrid entangled states, and one such commonly investigated state is the $|\psi\rangle_{AB} = (1/\sqrt{2})(|0\rangle_A|\alpha\rangle_B + |1\rangle_A|-\alpha\rangle_B)$ state, which shows bipartite hybrid entanglement between a qubit and a CV mode (commonly referred to as an entangled 'Schrödinger cat state'). This can be prepared in a multitude of ways, such as via use of polarization photons, probabilistic heralded single-photon measurements and Hadamard gates [53], by relying on Kerr nonlinearities [54–57], or by exploiting entangled polarization qubits with a series of beam-splitters (BSs) and auxiliary modes [58]. In fact, previous work has investigated this particular hybrid state in the same ES protocol as followed in this work, demonstrating that this simple hybrid state is theoretically resilient to low levels of photonic losses [59,60].

## (b) Stage 1: *Preparation of hybrid entanglement*

In this work, we consider hybrid entangled cat states, in which the CV components in the superposition are themselves cat states, as discussed in the previous section.

Alice's input quantum state to our proposed ES protocol is then described mathematically as follows

$$|\Psi\rangle_{AB}^{\text{HE}} = \frac{\mathcal{N}_\alpha}{\sqrt{2}}\big[|0\rangle_A\big(|\alpha\rangle_B + |-\alpha\rangle_B\big) + |1\rangle_A(|i\alpha\rangle_B + |-i\alpha\rangle_B)\big], \tag{2.3}$$

in which $\mathcal{N}_\alpha = 1/\sqrt{2 + 2\,e^{-2\alpha^2}}$ is the normalization of an even (or odd) cat state. Furthermore, Bob's hybrid state of modes $C$ and $D$ ($|\Psi\rangle_{CD}^{\text{HE}}$) is identical to that of Alice's in modes $A$ and $B$, as described in equation (2.3), and so our total quantum state at this stage is

$$|\Psi\rangle_{ABCD}^{\text{HE}} = |\Psi\rangle_{AB}^{\text{HE}} \otimes |\Psi\rangle_{CD}^{\text{HE}}. \tag{2.4}$$

The hybrid states used in the protocol discussed here have increased complexity and thus require additional preparation steps, compared to the usual hybrid states. We, therefore, propose a sequence of quantum operations capable of producing such complex hybrid states, as the first stage of our protocol, as follows:

1. Begin with an initial product state of

$$|\Psi\rangle_{AB} = \mathcal{N}_\alpha[|0\rangle_A(|\alpha\rangle_B + |-\alpha\rangle_B)]. \tag{2.5}$$

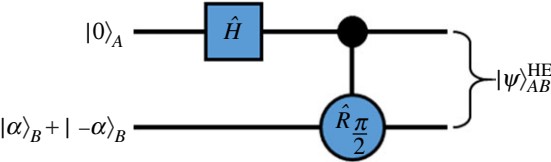

**Figure 2.** A quantum circuit which could be used to prepare the initial hybrid entangled states ($|\Psi\rangle_{AB}^{HE}$ and $|\Psi\rangle_{CD}^{HE}$, as given in equation (2.3)) for our proposed entanglement swapping protocol, using a Hadamard gate ($\hat{H}$) and a controlled $\pi/2$ rotation gate ($\hat{R}_{\pi/2}$). (Online version in colour.)

2. Apply a Hadamard gate ($\hat{H}$) to mode $A$, such that $|0\rangle_A \xrightarrow{\hat{H}} (1/\sqrt{2})(|0\rangle_A + |1\rangle_A)$. Although a Hadamard gate performed on a single mode Fock state to produce a superposition state is not trivial, it is still practically possible to perform, and a feasible experimental method is given in [61]. The state after this is then

$$|\Psi\rangle_{AB} \xrightarrow{\hat{H}} \frac{\mathcal{N}_\alpha}{\sqrt{2}}[|0\rangle_A(|\alpha\rangle_B + |-\alpha\rangle_B) + |1\rangle_A(|\alpha\rangle_B + |-\alpha\rangle_B)]. \tag{2.6}$$

3. Finally, perform a conditional (controlled) $\pi/2$ rotation on mode $B$, such that when the qubit in mode $A$ is $|1\rangle_A$ then this rotation is performed on mode $B$. This can be shown as

$$\mathcal{N}_\alpha(|\alpha\rangle_B + |-\alpha\rangle_B) \xrightarrow{\hat{R}_{\frac{\pi}{2}}} \mathcal{N}_\alpha|i\alpha\rangle_B + |-i\alpha\rangle_B. \tag{2.7}$$

The final hybrid entangled state produced from this sequence of quantum logic gates is then that of equation (2.3).

This process can be shown as a quantum circuit as per figure 2: This circuit, therefore, demonstrates that these particular hybrid states can be produced using standard quantum operations.

## (c) Stage 2: *Lossy optical modes*

In any practical demonstration of this protocol, there will be intrinsic photonic losses that occur in the propagating modes (such as within optical fibres), and so we model these losses to investigate the tolerance of our protocol to this (see Stage 2 of figure 1).

To model photonic losses, we combine the lossy modes ($B$ and $D$) with beam-splitters of transmission $T$, along with input vacuum states in modes $E_B$ and $E_D$ ($|0\rangle_{E_B}$ and $|0\rangle_{E_D}$), for losses in modes $B$ and $D$, respectively, and then trace out the loss modes as lost to the environment. Therefore, by decreasing the value of $T$ from unity, we introduce greater levels of photonic loss in the system.

For now, we will consider the case in which the two beam-splitters ($BS_{B,E_B}^{T_B}$ and $BS_{D,E_D}^{T_D}$) induce equal amounts of loss between modes $B$ and $D$, thus $T_B = T_D$. In §3, we discuss the more realistic circumstance in which $T_B \neq T_D$.

To demonstrate how we mathematically account for photonic losses, consider an arbitrary coherent state $|\beta\rangle_i$ of complex amplitude $\beta$ in mode $i$, which we combine with a BS of transmission $T_i$, along with a vacuum state in mode $j$

$$BS_{i,j}^{T_i}|\beta\rangle_i|0\rangle_j = \exp\left[\sqrt{T_i}\beta\hat{a}^\dagger - \sqrt{T_i}\beta^*\hat{a}\right] \times \exp\left[\sqrt{1-T_i}\beta\hat{b}^\dagger - \sqrt{1-T_i}\beta^*\hat{b}\right]|0\rangle_i|0\rangle_j$$

$$= \left|\sqrt{T_i}\beta\right\rangle_i\left|\sqrt{1-T_i}\beta\right\rangle_j, \tag{2.8}$$

where $\hat{a}^\dagger$ ($\hat{b}^\dagger$) and $\hat{a}$ ($\hat{b}$) are the creation and annihilation operators for modes $A$ and $B$, respectively.

The total quantum state of the ES protocol after inducing photonic losses is then described as

$$|\Psi\rangle_{ABE_BCDE_D} = \mathrm{BS}_{B,E_B}^{T_B}|\Psi\rangle_{AB}^{\mathrm{HE}}|0\rangle_{E_B} \otimes \mathrm{BS}_{D,E_D}^{T_D}|\Psi\rangle_{CD}^{\mathrm{HE}}|0\rangle_{E_D}, \tag{2.9}$$

in which,

$$|\Psi\rangle_{ABE_B} = \frac{(\mathcal{N}_\alpha)^2}{2} \times \Big[\Big(|0\rangle_A(|\sqrt{T_B}\alpha\rangle_B|\sqrt{1-T_B}\alpha\rangle_{E_B} + |-\sqrt{T_B}\alpha\rangle_B|-\sqrt{1-T_B}\alpha\rangle_{E_B})$$
$$+ |1\rangle_A(|\mathrm{i}\sqrt{T_B}\alpha\rangle_B|\mathrm{i}\sqrt{1-T_B}\alpha\rangle_{E_B} + |-\mathrm{i}\sqrt{T_B}\alpha\rangle_B|-\mathrm{i}\sqrt{1-T_B}\alpha\rangle_{E_B})\Big)\Big], \tag{2.10}$$

and the above state is identical to that describing modes $C$, $D$ and $E_D$.

## (d) Stage 3: *50 : 50 beam-splitter*

Following the introduction of loss into the propagating modes ($B$ and $D$), we then mix these modes at a 50 : 50 BS (as per Stage 3 of our protocol, outlined in figure 1). Consider now two arbitrary coherent states, of complex amplitudes $\alpha$ and $\eta$ in modes $i$ and $j$, respectively—the 50 : 50 BS ($\mathrm{BS}_{i,j}^{1/2}$) convention we use here can be shown mathematically as

$$\mathrm{BS}_{i,j}^{1/2}|\alpha\rangle_i|\eta\rangle_j = \left|\frac{\alpha-\eta}{\sqrt{2}}\right\rangle_i\left|\frac{\alpha+\eta}{\sqrt{2}}\right\rangle_j. \tag{2.11}$$

## (e) Stage 4: *Detection methods*

After mixing the lossy modes $B$ and $D$ via use of a 50 : 50 BS, we then need to detect these quantum states (Stage 4, figure 1) to project their respective wave-functions, thus ensuring successful ES to modes $A$ and $C$. We measure mode $B$ via a projective vacuum state measurement, and mode $D$ via balanced homodyne detection. For a detailed discussion of how we mathematically derive and implement both of these measurement techniques, refer to appendices A and B for the vacuum state detection and homodyne measurement schemes, respectively.

# 3. Modelling unequal photonic losses

In a practical setting, the two propagating modes $B$ and $D$ will not exhibit the exact same levels of photonic losses. For example, different lengths of optical fibres correspond to varying levels of loss—shorter fibre intrinsically exhibits lower losses. In fact, optical fibres of equal length may even exhibit differing photonic losses. There are also potential errors when coupling optical fibre to components, such as the 50 : 50 BS used to mix modes $B$ and $D$, or fibre splices within these modes. Hence, even if the lengths of fibre are identical, slightly different optical coupling in the two modes could give a small mismatch in losses. We could even consider the free-space ES scenario, in which the path lengths of modes $B$ and $D$ differ, and as such would exhibit varying levels of loss.

As such, the two BSs used to model this loss theoretically ($\mathrm{BS}_{B,E_B}^{T_B}$ and $\mathrm{BS}_{D,E_D}^{T_D}$) will not have equal transmission coefficients—that is to say that $T_B \neq T_D$. We therefore now determine whether allowing for a small difference in the losses experienced in these modes, which we denote $\upsilon$, impacts the quality of the entangled state shared between Alice and Bob post-protocol.

To avoid transmission coefficients exceeding unity (which is clearly unphysical), we parametrize our unequal loss parameter such that $T_B = T$ and $T_D = T - \upsilon$, in which $0 \leq T \leq 1$ and

$0 \leq \upsilon \leq 1$. When allowing for this 'loss mismatch', our total quantum state, after the lossy BSs, is then

$$
\begin{aligned}
|\Psi(\upsilon)\rangle_{ABE_BCDE_D} = \frac{(\mathcal{N}_\alpha)^2}{2} \times & \Big[ \Big( |0\rangle_A (|\sqrt{T}\alpha\rangle_B |\sqrt{\gamma}\alpha\rangle_{E_B} + |-\sqrt{T}\alpha\rangle_B |-\sqrt{\gamma}\alpha\rangle_{E_B}) \\
& + |1\rangle_A (|\mathrm{i}\sqrt{T}\alpha\rangle_B |\mathrm{i}\sqrt{\gamma}\alpha\rangle_{E_B} + |-\mathrm{i}\sqrt{T}\alpha\rangle_B |-\mathrm{i}\sqrt{\gamma}\alpha\rangle_{E_B}) \Big) \Big] \\
\otimes & \Big[ \Big( |0\rangle_C (|\sqrt{T-\upsilon}\alpha\rangle_D |\sqrt{\gamma+\upsilon}\alpha\rangle_{E_D} + |-\sqrt{T-\upsilon}\alpha\rangle_D |-\sqrt{\gamma+\upsilon}\alpha\rangle_{E_D}) \\
& + |1\rangle_C (|\mathrm{i}\sqrt{T-\upsilon}\alpha\rangle_D |\mathrm{i}\sqrt{\gamma+\upsilon}\alpha\rangle_{E_D} + |-\mathrm{i}\sqrt{T-\upsilon}\alpha\rangle_D |-\mathrm{i}\sqrt{\gamma+\upsilon}\alpha\rangle_{E_D}) \Big) \Big], \quad (3.1)
\end{aligned}
$$

where $\gamma = 1 - T$. Of course, in the limit of $\upsilon = 0$ we return to the equal loss scenario.

As opposed to selecting a specific value for this loss mismatch, it is logical to explore an average over $\upsilon$ by means of a one-sided (positive) Gaussian distribution, in which the width associated with this distribution corresponds to our *ensemble-averaged* loss mismatch value, which we denote $\Upsilon$. Considering this value as an ensemble average means that an experimentalist performing this ES protocol could have in mind a threshold of $\Upsilon$ for which they would know to not allow the mismatch in the loss to fall below.

The function for this Gaussian profile is calculated as

$$
f(\upsilon, \Upsilon) = \sqrt{\frac{2}{\pi \Upsilon^2}} \mathrm{e}^{-(\upsilon^2/2\Upsilon^2)}, \quad (3.2)
$$

which is normalized for $\int_0^\infty f(\upsilon, \Upsilon)\, \mathrm{d}\upsilon = 1$.

Therefore, the final state for our overall ES protocol, when accounting for unequal (averaged) losses between the propagating CV modes ($B$ and $D$), and after having performed vacuum state detection and a homodyne measurement, is

$$
\bar{\rho}_{AE_BCE_D}(\Upsilon) = \int_0^\infty f(\upsilon, \Upsilon)\rho_{AE_BCE_D}(\upsilon)\, \mathrm{d}\upsilon, \quad (3.3)
$$

where $\rho_{AE_BCE_D}(\upsilon) = |\Psi(\upsilon)\rangle_{AE_BCE_D}\langle\Psi(\upsilon)|$.

Lastly, we trace out the lossy modes ($E_B$ and $E_D$) as lost to the environment, using the coherent state to trace, thus giving our final density matrix as

$$
\mathrm{Tr}_{E_B, E_D}\left[\bar{\rho}_{AE_BCE_D}(\Upsilon)\right] = \bar{\rho}_{AC}(\Upsilon), \quad (3.4)
$$

which we use to determine all subsequent results presented throughout this work.

Due to the length of many of the mathematical expressions within this protocol, we do not include these in this work. We direct the reader to the work of [60] for an in-depth discussion, including all mathematical detail, of each step of this proposed protocol.

## 4. Results and discussion

Having performed successful ES, as per the protocol outlined in the prior sections, the state which Alice and Bob share (in the ideal, no loss, perfect measurements limits) is a phase-rotated Bell state

$$
|\Phi^+(\alpha)\rangle_{AC} = \frac{1}{\sqrt{2}} \left( \mathrm{e}^{-\mathrm{i}\alpha^2}|00\rangle_{AC} + \mathrm{e}^{+\mathrm{i}\alpha^2}|11\rangle_{AC} \right), \quad (4.1)
$$

and the orthogonal Bell state to this is

$$
|\Phi^-(\alpha)\rangle_{AC} = \frac{1}{\sqrt{2}} \left( \mathrm{e}^{-\mathrm{i}\alpha^2}|00\rangle_{AC} - \mathrm{e}^{+\mathrm{i}\alpha^2}|11\rangle_{AC} \right). \quad (4.2)
$$

This phase in the ideal outcome state (equation 4.1) could be corrected for via a suitable quantum operation (i.e. a phase-space rotation), or tracked through the protocol which the post-entangled state is being used for, such that the outcome state would be the maximally entangled $|\Phi^+\rangle_{AC} = (1/\sqrt{2})(|00\rangle_{AC} + |11\rangle_{AC})$ Bell state.

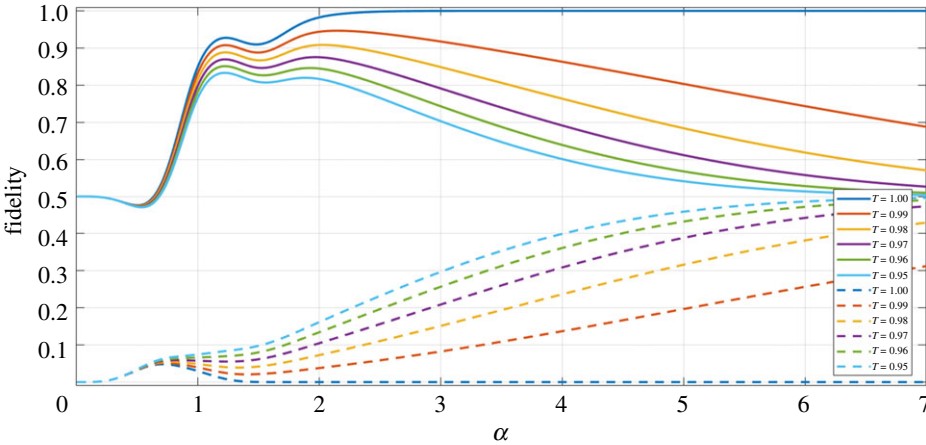

**Figure 3.** Fidelity against the $|\Phi^{+}(\alpha)\rangle$ (equation (4.1)) Bell state (solid lines), and the orthogonal $|\Phi^{-}(\alpha)\rangle$ (equation (4.2)) Bell state (dashed lines), as a function of the coherent state amplitude $\alpha$, for the final state generated via our entanglement swapping protocol (equation (3.4)), for varying levels of equal losses between modes $B$ and $D$. (Online version in colour.)

## (a) Fidelity

As the aim and purpose of our proposed ES protocol is to produce a specific Bell state of the highest quality (and therefore highest level of entanglement), the most useful measure of the quantum state shared between Alice and Bob is fidelity.

We calculate the fidelity ($F$) using the standard formula of

$$F = \langle \Phi^{+}(\alpha)|\bar{\rho}_{AC}(\Upsilon)|\Phi^{+}(\alpha)\rangle, \tag{4.3}$$

in which $|\Phi^{+}(\alpha)\rangle$ is the desired protocol outcome, and $\bar{\rho}_{AC}(\Upsilon)$ is the final density matrix of our ES protocol, given in equation (3.4). In the limit in which our protocol outcome is identical to $|\Phi^{+}(\alpha)\rangle$ the fidelity is $F = 1$, and as the closeness of these quantum states the fidelity drops from unity and approaches zero.

Figure 3 shows the fidelity against the phase-rotated Bell state, and the state orthogonal to this Bell state ($|\Phi^{-}(\alpha)\rangle$), as a function of the coherent state amplitude $\alpha$ of our final density matrix (equation (3.3)), but in the limit of $\Upsilon = 0$ (i.e. the losses experienced in modes $B$ and $D$ are equal).

Firstly, we can see in figure 3 that the plot of fidelity against the $|\Phi^{+}(\alpha)\rangle$ state plateaus at unity for $T = 1$ and $\alpha \geq 2.3$. This confirms that in the ideal limit (i.e. no loss) we can indeed produce the maximally entangled $|\Phi^{+}(\alpha)\rangle$ Bell state, following our ES protocol. What is also evident in figure 3 is that in the limit of very large $\alpha$, and for non-unity $T$ the fidelity of both plots (for $|\Phi^{+}(\alpha)\rangle$ and $|\Phi^{-}(\alpha)\rangle$) tends to $F = 0.50$; in fact, this 50 : 50 mixture of both Bell states is a mixed state, and exhibits no entanglement, and so is undesirable as the protocol outcome.

Moreover, we can also see in figure 3 that as we increase the level of losses in modes $B$ and $D$, the fidelity against the desired Bell state is lower for all $\alpha$—correspondingly, the fidelity therefore increases for the orthogonal Bell state, indicating the increased level of mixing.

If we consider the plot of figure 4, in which we now allow for an averaged loss mismatch between modes $B$ and $D$, we can see that increasing this loss mismatch value ($\Upsilon$) merely causes the fidelity plots to plateau to $F = 0.50$ more rapidly as a function of $\alpha$. In fact, this result is positive for the performance of this protocol—allowing for unequal losses between modes $B$ and $D$ does nothing more to impact the outcome of our protocol than simply increasing the level of equal losses between these modes.

Lastly, noticeable in both the equal loss and unequal loss plots (figures 3 and 4, respectively) is that there is a double peak present, for all values of $T$ and $\Upsilon$ investigated (although we recognize that the second peak becomes less pronounced as $T$ decreases and/or $\Upsilon$ increases).

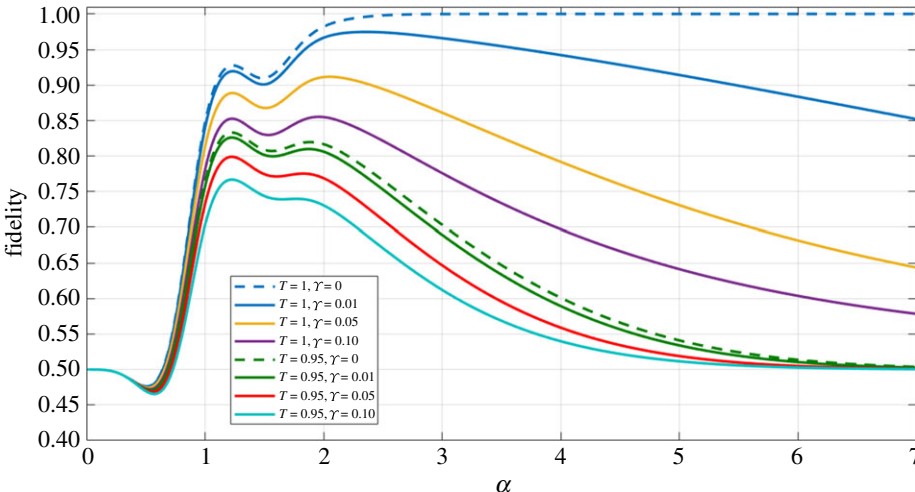

**Figure 4.** Fidelity against the $|\Phi^+(\alpha)\rangle$ (equation (4.1)) Bell state, as a function of the coherent state amplitude $\alpha$, for the final state generated via our entanglement swapping protocol (equation (3.4)), for varying levels of equal loss, and averaged unequal loss ($\Upsilon$). (Online version in colour.)

Mathematically, this is a direct consequence from the numerous exponential terms that are present in the final density matrix describing the resultant two-qubit matrix—we do not explicitly present this density matrix, as each of the matrix terms contains a vast number of complicated exponential terms, however, in appendix C we include the final quantum state (prior to tracing out the lossy modes), for the equal loss scenario. We also direct the reader to [60] for the mathematical detail of each stage of this ES protocol, including the final state generated.

These exponential terms present in the equal loss case of equation (A 1) (which are also present in the unequal loss scenario) can be seen as somewhat competing with each other; instead of seeing a simple peak followed by a decay due to exponential dampening, as a result of the introduction of loss into our system, we see a dip which is a consequence of exponential interference. This dip becomes less pronounced as the level of loss increases (for example, in the $T = 0.95$, $\Upsilon = 0.10$ plot of figure 4)—this is due to the exponential dampening effect, dependent on the level of loss, having a stronger impact on the final density matrix compared to the smaller effect of exponential interference.

Of course, this too is a positive result, as it means that we have a wide acceptance window of the coherent state amplitudes to prepare our initial states in, while still giving a tolerable fidelity. The maximum fidelity value varies as a function of $\alpha$, dependent on the level of loss considered: for $T = 0.97$ the second peak gives the maximum fidelity, and for $T \leq 0.96$, the first of these peaks gives the best outcome state, as shown in figure 3.

Finally, we also plot the fidelity against the desired $|\Phi^+(\alpha)\rangle$ Bell state, as a function of both $\alpha$ and the averaged unequal loss value $\Upsilon$, for $T = 1$, in figure 5.

The plot of figure 5 does not quite reach unity, even at the peak $\alpha$ value. The reason for this is that the numerical calculation cannot be evaluated for $\Upsilon$ very close to 0, as the Gaussian function then becomes a delta function (i.e. no longer a continuous spectrum, but instead is zero everywhere except for $\upsilon = 0$). Intrinsically, we expect that in the limit of $\Upsilon = 0$ we return to the results shown in the equal loss scenario (figure 3), and so we would indeed then expect the plot of figure 5 to reach, and plateau at, unity.

From figures 3 to 5, we can conclude that we do not desire an averaged loss mismatch value of $\Upsilon > 0.10$, as this gives an unacceptably low fidelity ($F \leq 0.80$) for all $\alpha$. We consider an acceptable fidelity result to be $F \geq 0.80$, as we could use any of the multitude of entanglement

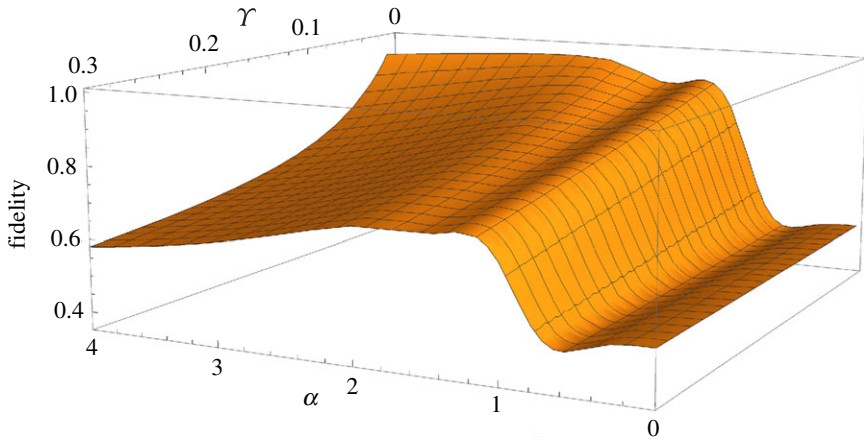

**Figure 5.** Fidelity against the $|\Phi^+(\alpha)\rangle$ (equation (4.1)) Bell state, as a function of the coherent state amplitude $\alpha$ and the averaged unequal loss value $\Upsilon$, for the final state generated via our entanglement swapping protocol (equation (3.4)), for $T = 1$. (Online version in colour.)

purification protocols available [62–66] to increase this fidelity to a sufficient level for further quantum communication/computation uses (i.e. to $F \geq 0.95$).

We acknowledge here that this limit we have for unequal losses $\Upsilon \leq 0.10$ to give an acceptable protocol output fidelity is not detrimental to the usefulness of our ES protocol: as previously stated, we refer to $\Upsilon$ as an average for unequal losses, as this variable is intended to reflect the practical perspective of running this protocol as an experiment, in which one would perhaps have a range of optical fibres, each of differing length, for example. It then follows that an averaged loss mismatch value of $\Upsilon > 0.10$ represents quite a large range of optical fibre lengths, and so it would be possible for an experimentalist to avoid unequal losses greater than this limit proposed in any case.

## (b) Imperfect homodyne detection

Thus far in the results and discussion, we have only analysed the scenario in which the homodyne measurement outcomes are the average *perfect* case. As detailed in appendix B, to allow for imperfections in the homodyne measurement outcome, we follow the method outlined in [67], and consider a resolution bandwidth ($\Delta x$) around the average measurement outcome. The final state of our protocol when analysing homodyne imperfections is given in appendix C §(a), as a density matrix in equation (C 4).

Analysis of homodyne measurement imperfections are vital within this work, as no practical homodyne detector is able to measure with a bandwidth equal to $\Delta x \approx 0$. Hence, in this section, we investigate the tolerance our protocol has to increasing this measurement bandwidth, while still producing a final state of acceptable fidelity. As we reduce $\Delta x$ we are effectively allowing for fewer possible measurement outcomes when performing the protocol practically, and this is how we determine success probability of the homodyne measurement, as discussed in §4(c).

Firstly, we plot the scenario of no loss (figure 6), to investigate the impact on output fidelity when increasing $\Delta x$ to investigate the impact in the most idealized circumstance. From figure 6, we can see that the fidelity against the ideal Bell state ($|\Phi^+(\alpha)\rangle$) begins to oscillate as we increase $\Delta x$. Intuitively, we can conclude that these oscillations are present as a result of the numerous competing exponentials, which are dependent on $\Delta x$ and also $\alpha$ within our final state (see equation (A 3)).

Interestingly, this oscillatory behaviour is not present when observing higher values of loss ($T = 0.95$) as per figure 7. This is because the dampening exponentials introduced when

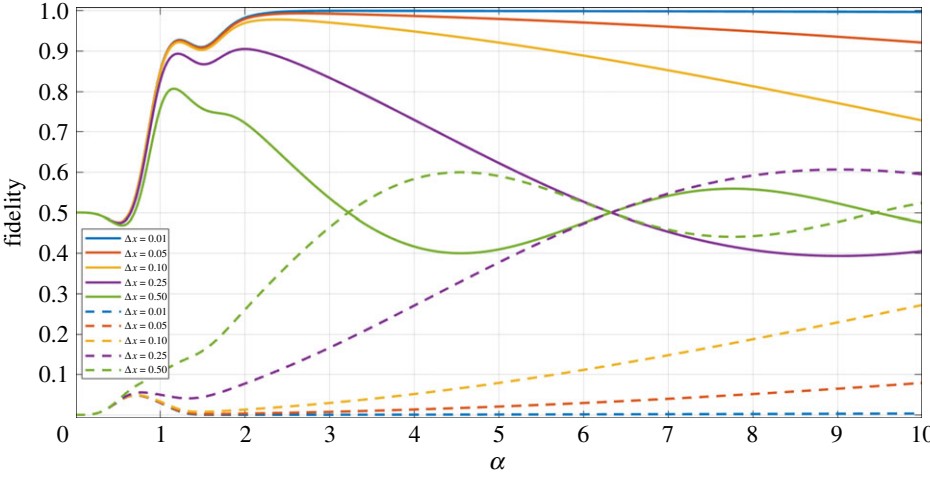

**Figure 6.** Fidelity against the $|\Phi^+(\alpha)\rangle$ (equation (4.1)) Bell state (solid lines in plot) and the orthogonal $|\Phi^-(\alpha)\rangle$ (equation (4.2)) Bell state (dotted lines in plot) as a function of $\alpha$ for the final state generated via our entanglement swapping protocol (equation (C 4)), for $T = 1$ and varying homodyne measurement bandwidth $\Delta x$. (Online version in colour.)

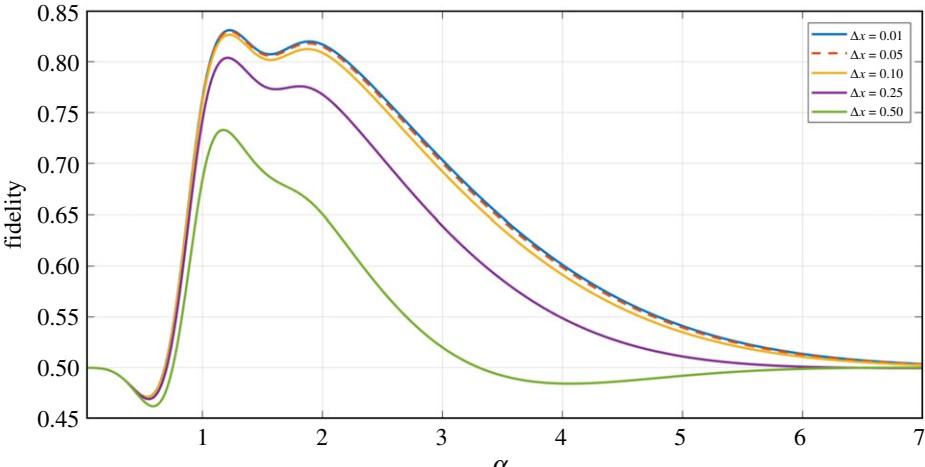

**Figure 7.** Fidelity against the $|\Phi^+(\alpha)\rangle$ (equation (4.1)) Bell state as a function of $\alpha$ for the final state generated via our cat state entanglement swapping protocol (equation (C 4)), for $T = 0.95$ and varying homodyne measurement bandwidth $\Delta x$. (Online version in colour.)

accounting for losses, after having traced out the lossy modes, have a stronger influence on the final density matrix compared to the competing exponentials which produce the oscillations.

Promisingly, we find that introducing a relatively high level of detection imperfection ($\Delta x \leq 0.25$) does not significantly impact the resultant state fidelity against the ideal Bell state, even in the higher loss scenario. In fact, it is still evident that there is a peak $\alpha$ value as noted in the previous fidelity plots discussed, and so if performing this practically one could take into account the known homodyne detector resolution bandwidth, and select the value of $\alpha$ which is likely to achieve the highest fidelity result.

Finally, we note that we do not show results for averaged unequal losses as these scale similarly to merely increasing the level of equal loss.

## (c) Homodyne measurement success probability

Another important quantity to consider, in evaluating the performance of a protocol, is the success probability of the measurement schemes. As previously discussed, evaluating the homodyne detection resolution bandwidth $\Delta x$ allows us to investigate the success probability of this measurement; increasing $\Delta x$ means allowing for more outcome results of this measurement, and so we expect to see the success probability increase as $\Delta x$ becomes larger.

Contrastingly, allowing for higher values of $\Delta x$ impacts the resultant fidelity of the output state of the protocol, as discussed in §4b—therefore, we expect to witness a trade-off between the success of the protocol and the quality (fidelity) of the outcome. This is useful to understand, because in some cases the customer might wish to suffer a lower success rate to obtain high fidelity pairs.

To calculate the homodyne success probability, we determine the modulus square of the normalized probability amplitudes (given by using the projector of equation (B 7) onto the coherent states present in mode $D$) to give us the probability distribution we integrate over, as

$$\left| {}_D \langle x_{\pi/4} | \Psi \rangle_{A\varepsilon_B CD\varepsilon_D} \right|^2, \tag{4.4}$$

which then gives the success probability as

$$\mathcal{P}_{\text{Hom.}}(\%)(\Upsilon) = \int_0^\infty f(\upsilon, \Upsilon) \mathcal{N}_\upsilon \left( \int_{(\mathcal{T}^+/2)\alpha - (\Delta x/2)}^{(\mathcal{T}^+/2)\alpha + (\Delta x/2)} \left| {}_D \langle x_{\pi/4} | \Psi \rangle_{A\varepsilon_B CD\varepsilon_D} \right|^2 \mathrm{d}x_{\pi/4} \right.$$
$$\left. + \int_{-(\mathcal{T}^+/2)\alpha - (\Delta x/2)}^{-(\mathcal{T}^+/2)\alpha + (\Delta x/2)} \left| {}_D \langle x_{\pi/4} | \Psi \rangle_{A\varepsilon_B CD\varepsilon_D} \right|^2 \mathrm{d}x_{\pi/4} \right) \mathrm{d}\upsilon \times 100, \tag{4.5}$$

where, $\mathcal{T}^+ = \sqrt{T} + \sqrt{T + \upsilon}$ and,

$$\mathcal{N}_\upsilon = \frac{1}{\left( 4 + 8\,\mathrm{e}^{-(|\mathcal{T}^+\alpha|^2)/4} + 24\,\mathrm{e}^{-(|\mathcal{T}^+\alpha|^2)/2} + 8\,\mathrm{e}^{-(|\mathcal{T}^+\alpha|^2)/4/3} + 4\,\mathrm{e}^{-|\mathcal{T}^+\alpha|^2} \atop + 8\,\mathrm{e}^{-(2+\mathrm{i})|\mathcal{T}^+\alpha|^2} + 8\,\mathrm{e}^{-(2-\mathrm{i})|\mathcal{T}^+\alpha|^2} \right)^{1/2}}, \tag{4.6}$$

is the normalization. Although we show the full expression above for clarity, we do not present the results for equal and averaged unequal losses for homodyne success probability, as these scale as expected—incremental increases in the level of loss (be that equal or unequal) slightly lowers the success probability.

Figure 8 shows the plot of the homodyne success probability $\mathcal{P}_{\text{Hom.}}(\%)$ as a function of $\alpha$, for increasing $\Delta x$. Here, we can see that for $\Delta x = 5.0$ the success probability is 100% for all $\alpha$—this is because the resolution bandwidth at such a high value covers the entire spectrum of the cat state probability distribution. To further understand the plot of figure 8 it is useful to look at the probability distribution of the cat state equation (equation (4.4)) for varying $\alpha$, as per figure 9.

In figure 9, $\alpha = 0$ gives the vacuum state probability distribution, as expected. Contrastingly, if we look to higher values of $\alpha$, then we see that for $\alpha = 2.0$ the two peaks of the probability distribution are almost entirely separated; this is the ideal scenario for successful homodyne detection, in which we have two outcome peaks to be measured (i.e. $x_{\pi/4} = \pm\alpha$—see figure 11).

Moreover, when considering $\alpha = 1.0$ in figure 9, we can see that the width of this peak is broader than that of $\alpha = 0$. This is as a result of the vacuum states present in mode $D$, which are not ideal states for the homodyne measurement to project to, as measurement of these will not give an entangled output. These vacuum states are exponentially dampened as a function of $\alpha$, and so it is not until $\alpha > 2.0$ when the contribution by the vacuum is reduced to a negligible amount.

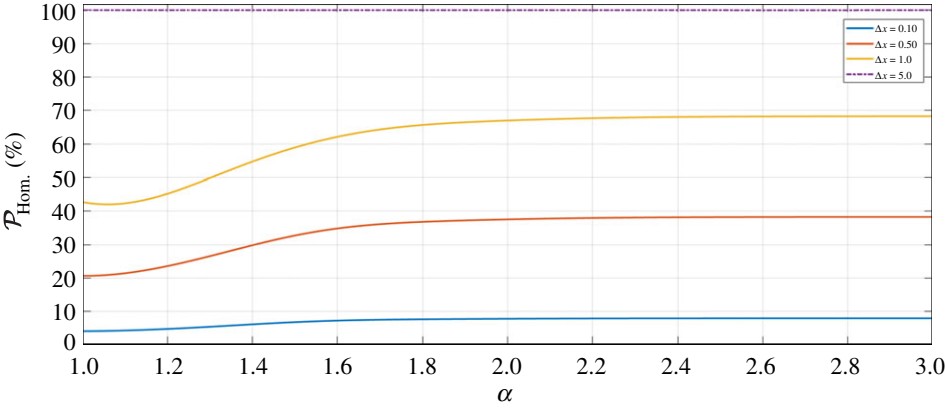

**Figure 8.** Success probability ($\mathcal{P}_{\text{Hom.}}$(%)) of the homodyne measurement (equation (4.5)), as a function of $\alpha$, for varying homodyne measurement bandwidth $\Delta x$ and $T = 1.0$. (Online version in colour.)

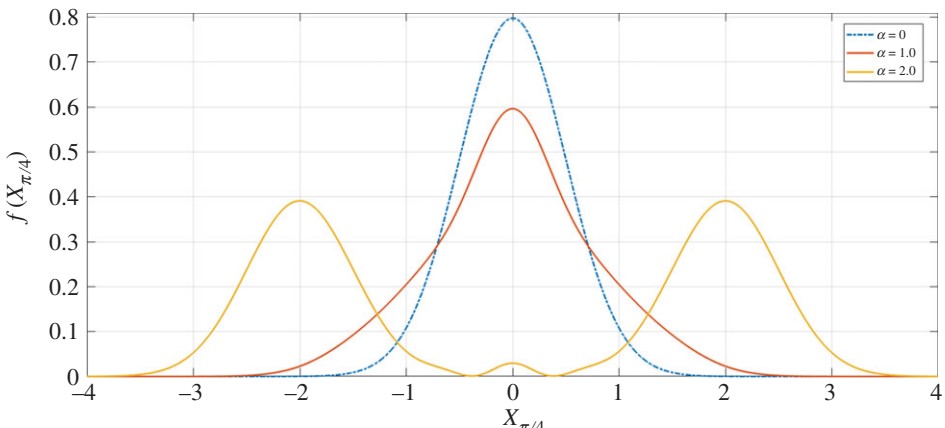

**Figure 9.** Probability distribution $f(x_{\pi/4})$ of the cat state equation (given by equation (4.4)), as a function of $x_{\pi/4}$, for $\alpha = 0$, $\alpha = 1.0$ and $\alpha = 2.0$ (for no loss). (Online version in colour.)

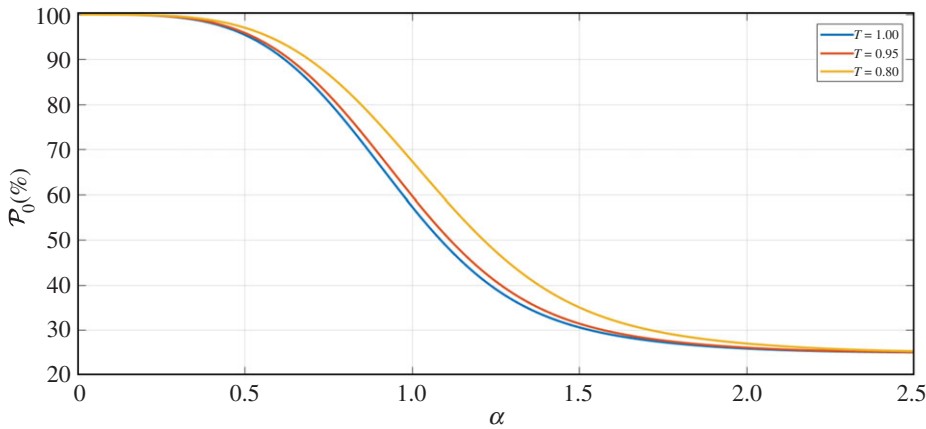

**Figure 10.** Success probability ($\mathcal{P}_0$(%)) of the vacuum measurement, as a function of the coherent state amplitude $\alpha$, for varying levels of equal losses between modes $B$ and $D$. Note that we truncate the $y$-axis of the plot so that the range is from 20% $\rightarrow$ 100%. (Online version in colour.)

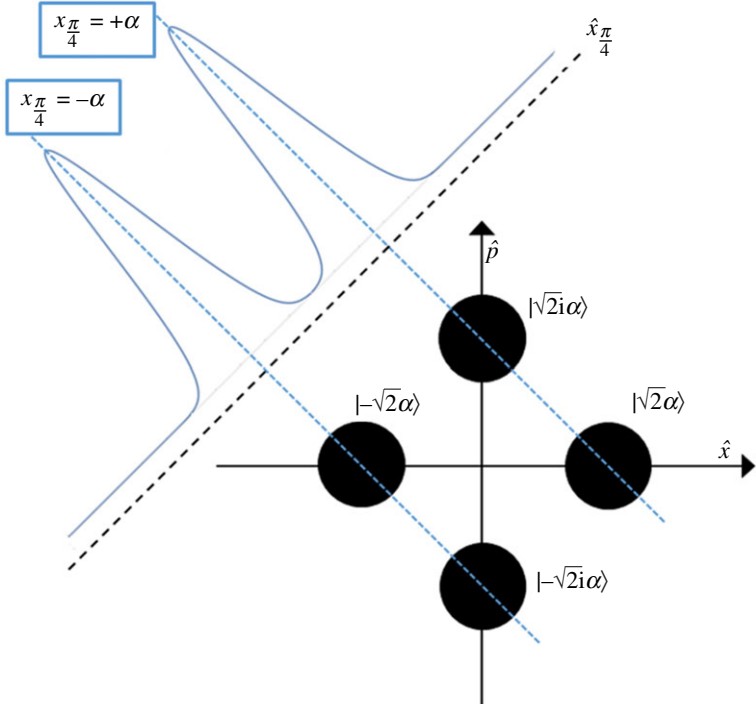

**Figure 11.** The position and momentum phase space occupied by two cat states $\mathcal{N}_\alpha(|\sqrt{2}\alpha\rangle + |-\sqrt{2}\alpha\rangle)$ and $\mathcal{N}_\alpha(|\sqrt{2}i\alpha\rangle + |-\sqrt{2}i\alpha\rangle)$, in which an $\hat{x}_{\pi/4}$ homodyne measurement yields a probability distribution exhibiting two peaks, centred at $x_{\pi/4} = \pm\alpha$. (Online version in colour.)

Finally, we highlight that we indeed witness a trade-off between homodyne success probability and the output state fidelity. The success probability is optimum for very large $\Delta x$, however, the output fidelity of the protocol would be very low in this scenario. Ideally, we would opt to use $\alpha$ values in the range of $1.0 \leq \alpha \leq 2.0$ and a measurement bandwidth of $\Delta x \leq 0.25$, which gives a success probability of around 10–15% in the no loss regime.

## (d) Vacuum measurement success probability

We now discuss the success probability associated with the vacuum state measurement. Figure 10 shows a plot of the vacuum measurement success probability ($\mathcal{P}_0$) as a function of the coherent state amplitude $\alpha$. We only plot up to $\alpha = 2.5$, as this is the region we are primarily concerned with in our protocol, due to the fact that this is where our fidelity values peak as a function of $\alpha$. Noticeably, we can see in figure 10 that for $\alpha = 0$, $\mathcal{P}_0 = 100\%$. Naturally, this is entirely expected, because at this value of $\alpha = 0$ all of these coherent states are in fact vacuum states, and so intrinsically any vacuum measurement here will always be performed with 100% success probability. More interestingly, we see that, for all values of loss covered in figure 10, all plots plateau at $\mathcal{P}_0 = 25\%$.

We also point out that although we have only considered equal losses between modes $B$ and $D$ in figure 10, the results are effectively identical when considering the same levels of unequal losses between these modes (as we noted in the fidelity plots previously discussed).

Given the fidelity results presented in figures 3–5, we can conclusively state that, for the levels of equal and unequal losses evaluated, we would desire coherent state amplitudes of $1.0 \leq \alpha \leq 2.0$,

so as to achieve the peak fidelity results. In this region, however, the success probability begins to drop, and in fact at the higher end of this limit ($\alpha \approx 2.0$), we can see in figure 10 that the success probability is around $\mathcal{P}_0 = 25\%$.

This trade-off between fidelity and success probability is a common occurrence in quantum communication schemes, and is something we must accept. Although success probability is undoubtedly significant when considering the practical implementation of our proposed ES protocol, we argue that high fidelity is far more important to aim for as opposed to better success probabilities; producing fewer pairs of higher fidelity quantum states is more useful for further quantum communication purposes, as opposed to producing a greater number of lower fidelity states.

Finally, it is worth addressing the point that even current state of the art single-photon detectors (SPDs) operate far from perfectly. There are various imperfections associated with SPDs, such as; photon number resolution, dead-time and recovery time, detector jitter, dark-counts, after-pulses and the detector efficiency itself [68]. Although all of these SPD imperfections are relevant to our set-up, as we use SPD to measure the presence of a vacuum state, the most important of these for our protocol that we should consider are detector efficiency and dark-counts. One such example of an off-the-shelf state of the art SPD is the ID Quantique ID281 SPD, which is based on superconducting nanowires [69], and is specified to achieve detector efficiencies of greater than or equal to 85% with dark count rates as low as 1–100 Hz. Moreover, such a detector is able to operate through optical fibres (as opposed to free-space), and also work over telecom wavelengths, which typically exhibit the lowest losses for propagating photons in optical fibre.

SPD imperfections, such as detector efficiency and dark-count rate, will clearly have a negative effect on the success probability of our protocol, and will therefore lower the vacuum measurement success probability further from the analysis already performed in this section. However, the overall output fidelity should not be impacted by these detector imperfections, and so to mitigate the impact we merely need to perform the protocol more times to achieve a successful outcome, thus lowering the protocol bit rate. We do note that while a lowering of the bit rate is not ideal, it is not too damaging for the overall usefulness of this protocol for its intended purpose; such an ES protocol as presented in this work is likely to be used to produce relatively low numbers of highly entangled pairs of photons for use in bespoke quantum networks, for example in disaggregated quantum computing networks, which will be discussed in the following section.

## 5. Application in a quantum network

Thus far, we have shown that one could theoretically produce a highly entangled Bell state of fidelity $F \geq 0.80$, when allowing for photonic transmissions of $T = 0.95$ in the propagating modes (corresponding to losses of $5\% \equiv 0.22$ dB). Currently, the highest-performing low-loss fibre available exhibits losses of $0.149$ dB km$^{-1}$ [70], and so 5% loss in our protocol corresponds to maximum distances between Alice and Bob of 3.0 km (assuming the measurement apparatus to detect modes $B$ and $D$ is located precisely in the middle of Alice and Bob), while still being able to share an entangled Bell state of fidelity $F \geq 0.80$.

Of course, this limits our protocol to being used in short-distance entanglement distribution networks. Nevertheless, this scheme could be highly suitable for sharing entanglement between adjacent quantum computers within a future quantum network, possibly in a local area network (LAN), such as a university campus or research centre.

However, if we allow for higher levels of loss, and therefore lower resultant fidelity, we can intrinsically distribute entanglement to two parties further distanced apart. If we set our fidelity acceptance threshold to $F \geq 0.70$, then the losses we can tolerate in modes $B$ and $D$ are $T = 0.88$ (equivalent to $0.56$ dB km$^{-1}$), allowing Alice and Bob to be located around 7.5 km apart if using ultra low-loss fibre of $0.149$ dB km$^{-1}$. Moreover, if we allow our fidelity acceptance window to drop even further to $F \geq 0.60$, then we can allow for losses of $T = 0.84$, which corresponds to

0.76 dB, thus allowing Alice and Bob to share an entangled state while separated by an overall distance of 10.5 km.

Again, this still makes our proposed entanglement swapping protocol suitably only for relatively short LAN-type distances, however, this comes at the advantage of being able to distribute highly entangled Bell states.

A Bell state of fidelity $F \leq 0.95$ is impractical for further uses with quantum computing, communications or information processing, and so allowing for further losses within our protocol means that a subsequent entanglement purification scheme would be required. Nevertheless, this is a common requirement for entanglement swapping protocols, and proposed quantum repeater networks use entanglement purification nodes as a fundamental part of the protocol.

Research into increasing the level of entanglement shared between two distant parties has been carried out extensively, and there exist a multitude of potential protocols which can increase Bell state fidelity from $F = 0.60$ to $F \geq 0.95$ [62–66]. Of course, in this circumstance this then requires one to use a higher number of lower fidelity pairs to produce one very high fidelity pair, although we argue that for the intended purpose of this protocol it is logical to sacrifice high bit rates to ensure that the entangled pairs delivered are of the best quality.

# 6. Conclusion

Following our proposed ES protocol, we can produce a phase-rotated Bell state $|\Phi^+(\alpha)\rangle$ between two distant nodes, with fidelity of $F \geq 0.80$ for equal losses between modes $B$ and $D$ of $T \geq 0.95$ with an averaged loss mismatch value of $\Upsilon \geq 0.05$. We find that introduction of averaged unequal losses between the propagating modes does not significantly impact the protocol results, and in fact has no more of an effect than merely increasing the levels of equal losses between these channels.

We witness a double peak in all fidelity plots, as a function of the coherent state amplitude $\alpha$, and so we have a broader range of acceptable $\alpha$ values than in our previously proposed (coherent state superposition) protocol of [59], in which we witness only a single sharp peak. The protocol investigated in this work is also slightly more tolerant to induced losses than the work of [59].

For the range of $\alpha$ which delivers a fidelity of $F \geq 0.80$, in the higher loss limits of ($T = 0.95$ and $\Upsilon = 0.05$) the success probability of the vacuum measurement is $\mathcal{P}_0 \approx 40\%$. We can allow for a fairly broad homodyne measurement bandwidth of $\Delta x \leq 0.50$, while still giving an entangled output state of respectable fidelity against the ideal case.

We reiterate that the phase present in the Bell state produced through our protocol, $|\Phi^+(\alpha)\rangle$, is not of detriment to the usefulness of this protocol in distributing entanglement to two parties; however, in a practical implementation, this would require the customer to be informed of the exact phase present each time. As this phase is fixed by the value of $\alpha$ chosen, all post-entangled pairs received by the two parties will have the same phase.

Moreover, we conclude that this protocol is tolerant only to low levels of photonic losses, and so is more suitable for distributing entangled quantum states over a local area network between parties located 5–10 km apart, when followed by a suitable entanglement purification scheme. These entangled pairs of photons could then be used for further quantum communication purposes, or by adjacent quantum computing processors, which may require entanglement for communications or information processing [15].

Data accessibility. The code used to generate the data presented in this research is available at Dryad under the name 'Mathematica code for cat state entanglement swapping' (doi:10.5061/dryad.05qfttf0c).

Authors' contributions. R.C.P. carried out this work as part of their PhD research project, supervised by T.P.S. and J.J. carried out the initial work on this project, before handing over to R.C.P., as well as the original source code. All authors gave final approval for publication and agree to be held accountable for the work performed therein.

Competing interests. We declare we have no competing interests.

Funding. We acknowledge support from EPSRC grant no. (EP/M013472/1)

Acknowledgements. The authors thank Prof. Andrew Lord for useful discussions.

# Appendix A. Vacuum state detection

By 'vacuum state measurement' we mean that to reveal the absence of a photon (therefore indicating a vacuum state) one could use a *perfect* photodetector, and upon not hearing the characteristic *click* of the detector, indicating the presence of one or more photons, it can be assumed that there is not a photon present [71].

To measure the presence of a vacuum, or lack thereof, we apply a positive-operator valued measure (POVM) described by the operator [72]

$$\hat{P}_i^0 = |0\rangle_i\langle 0|, \tag{A 1}$$

where $|0\rangle_i$ represents a vacuum state in mode $i$. This POVM measurement can be calculated by application of this vacuum projector $\hat{P}_i^0$ onto a coherent state (as is required in our proposed ES protocol). For example, consider an arbitrary coherent state of complex amplitude $\gamma$ in mode $i$ as follows:

$$|0\rangle_i\langle 0|\gamma\rangle_i = |0\rangle_i\, e^{-(|\gamma|^2/2)} \sum_{n=0}^{\infty} \frac{\gamma^n}{\sqrt{n!}}\,_i\langle 0|n\rangle_i$$

$$= |0\rangle_i\, e^{-(|\gamma|^2/2)} \sum_{n=0}^{\infty} \frac{\gamma^n}{\sqrt{n!}} \delta_{n,0} = |0\rangle_i\, e^{-(|\gamma|^2/2)}, \tag{A 2}$$

in which we have applied the Fock basis representation of the coherent state ($|\gamma\rangle_i = e^{-(|\gamma|^2/2)}$ $\sum_{n=0}^{\infty}(\gamma^n/\sqrt{n!})|n\rangle_i$), and we have made use of the Kronecker delta function, in which $\delta_{n,0} = 0$ for $n \neq 0$ and $\delta_{n,0} = 1$ for $n = 0$. We also highlight here that the application of this projective vacuum state measurement introduces an exponential dampening term of $e^{-(|\gamma|^2/2)}$, which is intrinsically important to the resultant performance of our proposed ES protocol, as will be discussed later.

After performing this projective vacuum measurement, our total quantum state is

$$\hat{P}_B^0|\Psi\rangle_{ABE_BCDE_D} = \sqrt{\mathcal{P}_0}|\Psi\rangle_{AE_BCDE_D}, \tag{A 3}$$

in which $\mathcal{P}_0$ is the success probability of the vacuum measurement (this will be discussed further in §(d)). For clarity, we omit the state $|0\rangle_B$ on the right-hand side of equation (A 3) (and in further expressions), as this is the remaining state from the vacuum measurement operator (equation (A 1)) after projecting mode $B$, and this will be used to project the complex conjugate of mode $B$ when used to form the final density matrix.

# Appendix B. Homodyne measurement

Following the vacuum measurement of mode $B$, we proceed to measure mode $D$ via balanced homodyne detection. To perform balanced homodyne detection, the probe mode (mode $B_1$) is mixed at a 50:50 BS with a strong coherent field $|\beta\, e^{i\theta}\rangle$, in which $\beta$ is real, in mode $B_2$ (also referred to commonly as the local oscillator) of equal frequency, and photodetection is used to measure the outputs of both modes $B_1$ and $B_2$ [73–75].

The intensity difference between the two photodetectors ($D_{B_1}$ and $D_{B_2}$) is then calculated using the two mode operator $\hat{I}_{\hat{B}_1-\hat{B}_2}$ as

$$\hat{I}_{\hat{B}_1-\hat{B}_2} = \hat{b}_1^\dagger\hat{b}_2 + \hat{b}_2^\dagger\hat{b}_1. \tag{B 1}$$

Setting the local oscillator mode to $\hat{b}_2 = \beta\, e^{i\theta}$ then yields the expectation value as

$$\left\langle \hat{b}_1^\dagger\hat{b}_2 + \hat{b}_2^\dagger\hat{b}_1 \right\rangle = 2\beta\left\langle \hat{x}_\theta \right\rangle, \tag{B 2}$$

for $\hat{x}_\theta = \frac{1}{2}(\hat{b}_1 e^{-i\theta} + \hat{b}_1^\dagger e^{i\theta})$ [76], in which the phase of the quadrature to be measured is given by the phase $\theta$ of the local oscillator [77]. We adjust the phase angle such that $\hat{x}_{\theta\to 0} = \hat{x}$ and $\hat{x}_{\theta\to\pi/2} = \hat{p}$,

thus giving the position and momentum operators, respectively, as

$$\hat{x} = \frac{1}{2}(\hat{a} + \hat{a}^\dagger) \quad \text{and} \quad \hat{p} = \frac{i}{2}(\hat{a}^\dagger - \hat{a}), \tag{B 3}$$

in which the coherent state expectation values are

$$\langle \hat{x} \rangle = \frac{1}{2}(\alpha + \alpha^*) \quad \text{and} \quad \langle \hat{p} \rangle = \frac{1}{2i}(\alpha^* - \alpha), \tag{B 4}$$

for $\alpha = \alpha_x + i\alpha_y$ and $\alpha^* = \alpha_x - i\alpha_y$.

Finally, the probability amplitude of a projective homodyne measurement on an arbitrary coherent state $|\alpha\, e^{i\phi}\rangle$ is determined by projecting with an $\hat{x}_\theta$ eigenstate, in which $\hat{x}_\theta |x_\theta\rangle = x_\theta |x_\theta\rangle$ [78]

$$\langle x_\theta | \alpha\, e^{i\varphi} \rangle = \frac{1}{2^{-(1/4)}\pi^{(1/4)}} \exp\left[ -(x_\theta)^2 + 2\, e^{i(\varphi-\theta)}\alpha x_\theta - \frac{1}{2}\, e^{2i(\varphi-\theta)}\alpha^2 - \frac{1}{2}\alpha^2 \right], \tag{B 5}$$

where the subscript on $x_\theta$ is indicative of the angle in which the homodyne measurement is performed, and this angle can be accurately chosen through the phase of the local oscillator. We recognize here that homodyne detection is a routine and very accurate measurement technique used widely in optics as a means of measuring phase-dependent quantum phenomena [79–82].

The outcome of a homodyne measurement is a value $x_\theta$ of the continuous quadrature variable $\hat{x}_\theta$, and the resultant homodyne measurement value is given by a probability distribution that comes from the modulus squared of the wave function (as given in equation (B 5)). For this work, we therefore define an *ideal* homodyne measurement as the case when the resultant homodyne measurement value is at the maximum of the probability distribution, as indicated by the position and momentum phase space diagram of figure 11 (note that the states in this diagram have amplitudes of $\sqrt{2}\alpha$, which is a result of the 50 : 50 BS operation prior to this measurement).

Therefore, upon performing the homodyne measurement we have the total quantum state of

$$\hat{\Pi}_{HD}|\Psi\rangle_{AE_BCDE_D} = |\Psi\rangle_{AE_BCE_D}, \tag{B 6}$$

in which $\hat{\Pi}_{HD} = |x_{\pi/4}\rangle_D\langle *|x_{\pi/4}$ is the homodyne measurement projector, of measurement angle $\pi/4$. Again, as with the measurement of mode $B$ (equation (A 3)), we omit mode $D$ on the right-hand side of this expression, as this is removed when forming the final state density matrix.

We select the angle of measurement $\theta = \pi/4$ such that we *quantum erase* information between certain peaks in the probability amplitude. To clarify, the purpose of the homodyne measurement in this scenario is to *indistinguishably* detect the coherent states, thus causing the output state to exhibit entanglement; the homodyne measurement outcome heralds the entangled state that is produced.

Considering the position and momentum of the cat states in figure 11, were we to measure along the $\hat{x}_{\pi/2}$ axis, for example, then we would only erase information between the two states along the momentum axis ($|\sqrt{2}i\alpha\rangle$ and $|-\sqrt{2}i\alpha\rangle$). This means that the output probability distribution would have three peaks, thus causing the remaining quantum state to be less entangled than for the circumstance in which we quantum erase information between the states by an $\hat{x}_{\pi/4}$ measurement, as shown in figure 11. Of course, ideally one would perform this measurement with an angle such that there is only a single peak in the resultant probability distribution, however, this is impossible for the case at hand.

A homodyne measurement along the $\hat{x}_{\pi/4}$ axis, as shown in figure 11 has two peaks centred at $x_{\pi/4} = \pm\alpha$, and so we use this result as our homodyne measurement outcome in establishing the final quantum state in this ES protocol. We emphasize that a homodyne measurement is a continuous quadrature measurement, and so realistically there is a range of outcome values in which $x_\theta$ may take (hence, as per figure 10, $x_{\pi/4} = \pm\alpha$ are the average *perfect* outcomes). Therefore, to investigate *imperfect* homodyne detection, we need to allow for a resolution bandwidth about the expected measurement outcome value. The mathematical method for this was derived in [67], and we use this in determining the acceptance value of how large the resolution bandwidth can be, while still producing entangled qubits of reasonable fidelity.

Following the derivation detailed in [67], the imperfect homodyne measurement operator becomes

$$\hat{\Pi}_{\mathrm{HD}}(x_0, \Delta x) = \int_{x_0-(\Delta x/2)}^{x_0+(\Delta x/2)} |x_\theta\rangle\langle x_\theta| \, dx_\theta, \tag{B 7}$$

in which $x_0$ is the expected measured value and $\Delta x$ is the resolution bandwidth around this measured value. Intuitively, in the limit of $\Delta x \to 0$ we should approach the perfect homodyne measurement scenario as before. This, in fact, is how we determine success probability for the homodyne measurement, however this will be discussed in detail in §4c.

For successful ES, as per this protocol, it is essential that one performs a vacuum state measurement on one propagating mode, and a homodyne measurement on the other—it was found that if two homodyne measurements, or two vacuum measurements, are performed on modes $B$ and $D$ then the resultant quantum state is a linear combination of all possible two-qubit states, which is a product state and therefore exhibits no entanglement, and as such is of little use for further quantum communication/computation purposes.

We also note that we post-select the state prior to this measurement, conditional on the required vacuum projection outcome on mode $B$. That is to say, if we hear the photon detector *click* then we do not perform homodyne detection on mode $D$, but instead restart the entanglement swapping protocol again.

# Appendix C. The final state of the ES protocol

The final state (prior to tracing out the lossy modes $E_B$ and $E_D$) shared between Alice and Bob, for the scenario of equal losses between modes $B$ and $D$, is

$$
\begin{aligned}
|\Psi\rangle_{AE_BCE_D} = \mathcal{N}\Big[ |00\rangle_{AC}\Big( & \exp\big[(1-\mathrm{i})T\alpha^2\big]|\gamma\alpha\rangle_{E_B}|\gamma\alpha\rangle_{E_D} + \exp\big[-(3-3\mathrm{i})T\alpha^2\big]|-\gamma\alpha\rangle_{E_B}|-\gamma\alpha\rangle_{E_D} \\
& + \mathrm{e}^{-T\alpha^2}\big(|\gamma\alpha\rangle_{E_B}|-\gamma\alpha\rangle_{E_D} + |-\gamma\alpha\rangle_{E_B}|\gamma\alpha\rangle_{E_D}\big)\Big) \\
& + \mathrm{e}^{-T\alpha^2/2}|01\rangle_{AC}\Big( \exp\big[T\alpha^2\big]|\gamma\alpha\rangle_{E_B}|\gamma\mathrm{i}\alpha\rangle_{E_D} + \exp\big[-2T\mathrm{i}\alpha^2\big]|\gamma\alpha\rangle_{E_B}|-\gamma\mathrm{i}\alpha\rangle_{E_D} \\
& + \exp\big[2T\mathrm{i}\alpha^2\big]|-\gamma\alpha\rangle_{E_B}|\gamma\mathrm{i}\alpha\rangle_{E_D} + \exp\big[-3T\alpha^2\big]|-\gamma\alpha\rangle_{E_B}|-\gamma\mathrm{i}\alpha\rangle_{E_D} \\
& + \mathrm{e}^{-T\alpha^2/2}|10\rangle_{AC}\Big( \exp\big[T\alpha^2\big]|\gamma\mathrm{i}\alpha\rangle_{E_B}|\gamma\alpha\rangle_{E_D} + \exp\big[2T\mathrm{i}\alpha^2\big]|\gamma\mathrm{i}\alpha\rangle_{E_B}|-\gamma\alpha\rangle_{E_D} \\
& + \exp\big[-2T\mathrm{i}\alpha^2\big]|-\gamma\mathrm{i}\alpha\rangle_{E_B}|\gamma\alpha\rangle_{E_D} + \exp\big[-3T\alpha^2\big]|-\gamma\mathrm{i}\alpha\rangle_{E_B}|-\gamma\alpha\rangle_{E_D} \\
& + |11\rangle_{AC}\Big( \exp\big[(\mathrm{i}+1)T\alpha^2\big]|\gamma\mathrm{i}\alpha\rangle_{E_B}|\gamma\mathrm{i}\alpha\rangle_{E_D} + \exp\big[-(3+3\mathrm{i})T\alpha^2\big]|-\gamma\mathrm{i}\alpha\rangle_{E_B}|-\gamma\mathrm{i}\alpha\rangle_{E_D} \\
& + \mathrm{e}^{-T\alpha^2}\big(|\gamma\mathrm{i}\alpha\rangle_{E_B}|-\gamma\mathrm{i}\alpha\rangle_{E_D} + |-\gamma\mathrm{i}\alpha\rangle_{E_B}|\gamma\mathrm{i}\alpha\rangle_{E_D}\big)\Big)\Big],
\end{aligned}
\tag{C 1}
$$

in which $\mathcal{N}$ is the normalization coefficient, $\gamma = \sqrt{1-T}$, and $\alpha$ is real. In the limit of large $\alpha$ (i.e. $\alpha > 2.5$), and no loss ($T=1$), equation (A 1) reduces to the ideal Bell state outcome of $|\Phi^+(\alpha)\rangle = 1/\sqrt{2}(\mathrm{e}^{-\mathrm{i}\alpha^2}|00\rangle + \mathrm{e}^{+\mathrm{i}\alpha^2}|11\rangle)$.

We note here that we do not explicitly include the equivalent final quantum state which includes averaged unequal losses between modes $B$ and $D$, and also do not include the density matrix in which the lossy modes $E_B$ and $E_D$ are traced out, due to the length of these expressions. We point the reader to the work of [60] for the mathematical detail.

## (a) Final state with imperfect homodyne measurements

To derive the density matrix containing the homodyne bandwidth variable $\Delta x$, we begin with the equal loss state, immediately after the vacuum measurement and before the homodyne

measurement, given by equation (A 3). We then apply the imperfect homodyne operator as per equation (B 7), for measurement angle $\theta = \pi/4$, such that

$$
\begin{aligned}
|\Psi\rangle_{A\varepsilon_B C D\varepsilon_D} = \mathcal{N} \times \Big[ &|00\rangle_{AC}\Big[|\sqrt{2T}\alpha\rangle_D|\gamma\alpha\rangle_{\varepsilon_B}|\gamma\alpha\rangle_{\varepsilon_D} + |-\sqrt{2T}\alpha\rangle_D|-\gamma\alpha\rangle_{\varepsilon_B}|-\gamma\alpha\rangle_{\varepsilon_D} \\
&+ \mathrm{e}^{-T\alpha^2}|0\rangle_D\Big(|\gamma\alpha\rangle_{\varepsilon_B}|-\gamma\alpha\rangle_{\varepsilon_D} + |-\gamma\alpha\rangle_{\varepsilon_B}|\gamma\alpha\rangle_{\varepsilon_D}\Big)\Big] \\
&+ \mathrm{e}^{-T\alpha^2/2}|01\rangle_{AC}\Big[|\sqrt{T}\alpha\,\mathrm{e}^{\mathrm{i}\pi/4}\rangle_D|\gamma\alpha\rangle_{\varepsilon_B}|\gamma\mathrm{i}\alpha\rangle_{\varepsilon_D} + |\sqrt{T}\alpha\,\mathrm{e}^{-\mathrm{i}\pi/4}\rangle_D|\gamma\alpha\rangle_{\varepsilon_B}|-\gamma\mathrm{i}\alpha\rangle_{\varepsilon_D} \\
&+ |-\sqrt{T}\alpha\,\mathrm{e}^{-\mathrm{i}\pi/4}\rangle_D|-\gamma\alpha\rangle_{\varepsilon_B}|\gamma\mathrm{i}\alpha\rangle_{\varepsilon_D} + |-\sqrt{T}\alpha\,\mathrm{e}^{\mathrm{i}\pi/4}\rangle_D|-\gamma\alpha\rangle_{\varepsilon_B}|-\gamma\mathrm{i}\alpha\rangle_{\varepsilon_D}\Big] \\
&+ \mathrm{e}^{-T\alpha^2/2}|10\rangle_{AC}\Big[|\sqrt{T}\alpha\,\mathrm{e}^{\mathrm{i}\pi/4}\rangle_D|\gamma\mathrm{i}\alpha\rangle_{\varepsilon_B}|\gamma\alpha\rangle_{\varepsilon_D} + |-\sqrt{T}\alpha\,\mathrm{e}^{-\mathrm{i}\pi/4}\rangle_D|\gamma\mathrm{i}\alpha\rangle_{\varepsilon_B}|-\gamma\alpha\rangle_{\varepsilon_D} \\
&+ |\sqrt{T}\alpha\,\mathrm{e}^{-\mathrm{i}\pi/4}\rangle_D|-\gamma\mathrm{i}\alpha\rangle_{\varepsilon_B}|\gamma\alpha\rangle_{\varepsilon_D} + |-\sqrt{T}\alpha\,\mathrm{e}^{\mathrm{i}\pi/4}\rangle_D|-\gamma\mathrm{i}\alpha\rangle_{\varepsilon_B}|-\gamma\alpha\rangle_{\varepsilon_D}\Big] \\
&+ |11\rangle_{AC}\Big[|\sqrt{2T}\mathrm{i}\alpha\rangle_D|\gamma\mathrm{i}\alpha\rangle_{\varepsilon_B}|\gamma\mathrm{i}\alpha\rangle_{\varepsilon_D} + |-\sqrt{2T}\mathrm{i}\alpha\rangle_D|-\gamma\mathrm{i}\alpha\rangle_{\varepsilon_B}|-\gamma\mathrm{i}\alpha\rangle_{\varepsilon_D} \\
&+ \mathrm{e}^{-T\alpha^2}|0\rangle_D\Big(|\gamma\mathrm{i}\alpha\rangle_{\varepsilon_B}|-\gamma\mathrm{i}\alpha\rangle_{\varepsilon_D} + |-\gamma\mathrm{i}\alpha\rangle_{\varepsilon_B}|\gamma\mathrm{i}\alpha\rangle_{\varepsilon_D}\Big)\Big]\Big],
\end{aligned}
\tag{C 2}
$$

$$
\begin{aligned}
\int_{x_0-(\Delta x/2)}^{x_0+(\Delta x/2)} &|x_{\pi/4}\rangle_D\langle x_{\pi/4}|\Psi\rangle_{A\varepsilon_B C D\varepsilon_D}\,\mathrm{d}x_{\pi/4} \\
&= \int_{\pm\sqrt{T}\alpha-(\Delta x/2)}^{\pm\sqrt{T}\alpha+(\Delta x/2)} |\Psi\rangle_{A\varepsilon_B C\varepsilon_D}\,\mathrm{d}x_{\pi/4} = |\Psi(\Delta x)\rangle_{A\varepsilon_B C\varepsilon_D} \\
&= \int_{\pm\sqrt{T}\alpha-(\Delta x/2)}^{\pm\sqrt{T}\alpha+(\Delta x/2)} \mathcal{N}\frac{\exp\left[-(x_{\pi/4})^2\right]}{2^{-(1/2)}\pi^{(1/4)}} \\
&\quad\times \Big[|00\rangle_{AC}\Big(\exp\left[(1-\mathrm{i})2\sqrt{T}\alpha x_{\pi/4} + (\mathrm{i}-1)T\alpha^2\right]|\gamma\alpha\rangle_{\varepsilon_B}|\gamma\alpha\rangle_{\varepsilon_D} \\
&\quad+ \exp\left[-(1-\mathrm{i})2\sqrt{T}\alpha x_{\pi/4} + (\mathrm{i}-1)T\alpha^2\right]|-\gamma\alpha\rangle_{\varepsilon_B}|-\gamma\alpha\rangle_{\varepsilon_D} \\
&\quad+ \mathrm{e}^{-T\alpha^2}\Big(|\gamma\alpha\rangle_{\varepsilon_B}|-\gamma\alpha\rangle_{\varepsilon_D} + |-\gamma\alpha\rangle_{\varepsilon_B}|\gamma\alpha\rangle_{\varepsilon_D}\Big)\Big) \\
&\quad+ \mathrm{e}^{-T\alpha^2/2}|01\rangle_{AC}\Big(\exp\left[2\sqrt{T}\alpha x_{\pi/4} - T\alpha^2\right]|\gamma\alpha\rangle_{\varepsilon_B}|\gamma\mathrm{i}\alpha\rangle_{\varepsilon_D} + \exp\left[-2\sqrt{T}\mathrm{i}\alpha x_{\pi/4}\right]|\gamma\alpha\rangle_{\varepsilon_B}|-\gamma\mathrm{i}\alpha\rangle_{\varepsilon_D} \\
&\quad+ \exp\left[2\sqrt{T}\mathrm{i}\alpha x_{\pi/4}\right]|-\gamma\alpha\rangle_{\varepsilon_B}|\gamma\mathrm{i}\alpha\rangle_{\varepsilon_D} + \exp\left[-2\sqrt{T}\alpha x_{\pi/4} - T\alpha^2\right]|-\gamma\alpha\rangle_{\varepsilon_B}|-\gamma\mathrm{i}\alpha\rangle_{\varepsilon_D}\Big) \\
&\quad+ \mathrm{e}^{-T\alpha^2/2}|10\rangle_{AC}\Big(\exp\left[2\sqrt{T}\alpha x_{\pi/4} - T\alpha^2\right]|\gamma\mathrm{i}\alpha\rangle_{\varepsilon_B}|\gamma\alpha\rangle_{\varepsilon_D} + \exp\left[2\sqrt{T}\mathrm{i}\alpha x_{\pi/4}\right]|\gamma\mathrm{i}\alpha\rangle_{\varepsilon_B}|-\gamma\alpha\rangle_{\varepsilon_D} \\
&\quad+ \exp\left[-2\sqrt{T}\mathrm{i}\alpha x_{\pi/4}\right]|-\gamma\mathrm{i}\alpha\rangle_{\varepsilon_B}|\gamma\alpha\rangle_{\varepsilon_D} + \exp\left[-2\sqrt{T}\alpha x_{\pi/4} - T\alpha^2\right]|-\gamma\mathrm{i}\alpha\rangle_{\varepsilon_B}|-\gamma\alpha\rangle_{\varepsilon_D}\Big) \\
&\quad+ |11\rangle_{AC}\Big(\exp\left[(\mathrm{i}+1)2\sqrt{T}\alpha x_{\pi/4} - (\mathrm{i}+1)T\alpha^2\right]|\gamma\mathrm{i}\alpha\rangle_{\varepsilon_B}|\gamma\mathrm{i}\alpha\rangle_{\varepsilon_D} \\
&\quad+ \exp\left[-(1+\mathrm{i})2\sqrt{T}\alpha x_{\pi/4} - (\mathrm{i}+1)T\alpha^2\right]|-\gamma\mathrm{i}\alpha\rangle_{\varepsilon_B}|-\gamma\mathrm{i}\alpha\rangle_{\varepsilon_D} \\
&\quad+ \mathrm{e}^{-T\alpha^2}\Big(|\gamma\mathrm{i}\alpha\rangle_{\varepsilon_B}|-\gamma\mathrm{i}\alpha\rangle_{\varepsilon_D} + |-\gamma\mathrm{i}\alpha\rangle_{\varepsilon_B}|\gamma\mathrm{i}\alpha\rangle_{\varepsilon_D}\Big)\Big)\Big]\mathrm{d}x_{\pi/4},
\end{aligned}
\tag{C 3}
$$

in which we have set $x_0 = \pm\sqrt{T}\alpha$ in equation (A 3), as these are the ideal homodyne outcomes. Finally, to construct the final density matrix we merely trace out the lossy modes as before, giving the final state as

$$
\rho_{AC}(\Delta x) = \mathrm{Tr}_{\varepsilon_B,\varepsilon_D}\left[\int_{\pm\sqrt{T}\alpha-(\Delta x/2)}^{\pm\sqrt{T}\alpha+(\Delta x/2)} |\Psi(\Delta x)\rangle_{A\varepsilon_B C\varepsilon_D}\langle\Psi(\Delta x)|\,\mathrm{d}x_{\pi/4}\right].
\tag{C 4}
$$

Note that we do not include the equivalent expressions for unequal losses and homodyne imperfections, due to the length of the derivation—however, we again point the reader to [60] for the detail.

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
