## [Reviewer comments · Proceedings. Mathematical, Physical, and Engineering Sciences]

Review History

RSPA-2020-0237.R0 (Original submission)

Review form: Referee 1

Is the manuscript an original and important contribution to its field?

Marginal

Is the paper of sufficient general interest?

Acceptable

Is the overall quality of the paper suitable?

Acceptable

Can the paper be shortened without overall detriment to the main message?

No

Do you think some of the material would be more appropriate as an electronic appendix?

No

Do you have any ethical concerns with this paper?

No

Recommendation?

Major revision is needed (please make suggestions in comments)

Comments to the Author(s)

The authors of the submitted manuscript suggest a scheme for entanglement swapping between hybrid entangled states. The scheme itself was already suggested in the authors' previous work (reference [46]). In the present work, a different type of hybrid entangled states are considered and some advantages of using them are discussed. To my opinion, there are some points that should be addressed by the authors with appropriate revision before the paper can be considered for publication in the journal.

- Preparation of hybrid entanglement (Section 2.2) is not practical. The kind of qubit that the authors have in mind seems that of Fock states. However, if so, It will be highly nontrivial to realize a Hadamard gate. The authors should discuss how to perform such an operation with its limitations.
- In the proposed scheme, the preparation of hybrid entanglement requires cat states. The authors must then discuss how to prepare cat states first with appropriate references and current experimental limitations.
- The authors consider imperfect homodyne detection, which is reasonable. However, the vacuum measurement used for this scheme could be experimentally even more challenging due to the limited efficiency of the detector. This issue should be discussed and taken into account properly in the context.
- In page 17, an explicit expression of the final state of the ES protocol, prior to tracing out the lossy modes, is presented. I believe that the reader would be more interested in the expression after tracing out the lossy modes. I suggest that the author present the expression after tracing out.
- The authors should add some more references, for example, Sheng et al.'s work on hybrid entanglement purification (PRA 88, 022302, 2013) as well as some earlier works on cat-state preparation (e.g. Ourjoumtsev et al., Nature 448, 784, 2007).
- In addition, presentation of the manuscript need to be improved.
- * There are some textbook-type contents such as 2.5.2. throughout the paper that could be shortened or moved to Appendix.
- * The abstract would look more appropriate if it is in one paragraph.
- * There are too many figures. The authors should consider whether all the figures are necessary. For example, I wonder if the low-fidelity plots (e.g. Fig. 12 and dotted curves in Fig. 10 and Fig. 7) would be of any interest to the reader. They might be dropped, and some of the figures might be put together under one caption.

The manuscript would be publishable in the journal after such revision is properly made.

Review form: Referee 2

Is the manuscript an original and important contribution to its field?

Good

Is the paper of sufficient general interest?

Good

Is the overall quality of the paper suitable?

Good

Can the paper be shortened without overall detriment to the main message?

Yes

Do you think some of the material would be more appropriate as an electronic appendix?

No

Do you have any ethical concerns with this paper?

No

Recommendation?

Accept with minor revision (please list in comments)

Comments to the Author(s)

The authors have addressed the problem of entanglement Swapping for potential application of quantum networking for a distance of 5-10 km using a discrete variable (Fock state) and a continuous variable (cat state superposition) entangled state. The topic looks interesting and the results produced are well explained to understand the problem. The authors concluded that their suggested protocol is tolerant only to low levels of photonic losses. Acknowledging the contributions of the paper the reviewer has the following points which require attention or more explanation from the authors.

1. In the introduction section of the paper, more references are needed to highlight existing ES techniques along with their merits and demerits.
2. How the proposed Scheme overcome disadvantages of the previous schemes? (if there is any) Please clearly describe your contributions against those findings.
3. Its better to write conclusion in paragraph instead of bullets.
4. Is it possible to generate any level of entanglement using the proposed entanglement swapping protocol or it will only generate a state of some fixed degree of entanglement?

Decision letter (RSPA-2020-0237.R0)

11-Aug-2020

Dear Dr Parker

The Editor of Proceedings A has now received comments from referees on the above paper and would like you to revise it in accordance with their suggestions which can be found below (not including confidential reports to the Editor).

Please submit a copy of your revised paper within four weeks - if we do not hear from you within this time then it will be assumed that the paper has been withdrawn. In exceptional circumstances, extensions may be possible if agreed with the Editorial Office in advance.

Please note that it is the editorial policy of Proceedings A to offer authors one round of revision in which to address changes requested by referees. If the revisions are not considered satisfactory by the Editor, then the paper will be rejected, and not considered further for publication by the journal. In the event that the author chooses not to address a referee's comments, and no scientific justification is included in their cover letter for this omission, it is at the discretion of the Editor whether to continue considering the manuscript.

- Acknowledgements
- Funding statement

To revise your manuscript, log into <http://mc.manuscriptcentral.com/prsa> and enter your Author Centre, where you will find your manuscript title listed under "Manuscripts with Decisions." Under "Actions," click on "Create a Revision." Your manuscript number has been appended to denote a revision.

You will be unable to make your revisions on the originally submitted version of the manuscript. Instead, revise your manuscript and upload a new version through your Author Centre.

When submitting your revised manuscript, you will be able to respond to the comments made by the referee(s) and upload a file "Response to Referees" in "Section 6 - File Upload". Please use this to document how you have responded to the comments, and the adjustments you have made. In order to expedite the processing of the revised manuscript, please be as specific as possible in your response to the referee(s).

IMPORTANT: Your original files are available to you when you upload your revised manuscript. Please delete any unnecessary previous files before uploading your revised version.

When revising your paper please ensure that it remains under 28 pages long. In addition, any pages over 20 will be subject to a charge (£150 + VAT (where applicable) per page). Your paper has been ESTIMATED to be 25 pages.

Once again, thank you for submitting your manuscript to Proc. R. Soc. A and I look forward to receiving your revision. If you have any questions at all, please do not hesitate to get in touch.

Yours sincerely
 Raminder Shergill
proceedingsa@royalsociety.org
 on behalf of
 Dr Ben Allen
 Board Member
 Proceedings A

Reviewer(s)' Comments to Author:

Referee: 1

Comments to the Author(s)

The authors of the submitted manuscript suggest a scheme for entanglement swapping between hybrid entangled states. The scheme itself was already suggested in the authors' previous work (reference [46]). In the present work, a different type of hybrid entangled states are considered and some advantages of using them are discussed. To my opinion, there are some points that should be addressed by the authors with appropriate revision before the paper can be considered for publication in the journal.

- Preparation of hybrid entanglement (Section 2.2) is not practical. The kind of qubit that the authors have in mind seems that of Fock states. However, if so, It will be highly nontrivial to realize a Hadamard gate. The authors should discuss how to perform such an operation with its limitations.
- In the proposed scheme, the preparation of hybrid entanglement requires cat states. The authors must then discuss how to prepare cat states first with appropriate references and current experimental limitations.
- The authors consider imperfect homodyne detection, which is reasonable. However, the vacuum measurement used for this scheme could be experimentally even more challenging due to the limited efficiency of the detector. This issue should be discussed and taken into account properly in the context.
- In page 17, an explicit expression of the final state of the ES protocol, prior to tracing out the lossy modes, is presented. I believe that the reader would be more interested in the expression

after tracing out the lossy modes. I suggest that the author present the expression after tracing out.

- The authors should add some more references, for example, Sheng et al.'s work on hybrid entanglement purification (PRA 88, 022302, 2013) as well as some earlier works on cat-state preparation (e.g. Ourjoumtsev et al., Nature 448, 784, 2007).

- In addition, presentation of the manuscript need to be improved.

* There are some textbook-type contents such as 2.5.2. throughout the paper that could be shortened or moved to Appendix.

* The abstract would look more appropriate if it is in one paragraph.

* There are too many figures. The authors should consider whether all the figures are necessary. For example, I wonder if the low-fidelity plots (e.g. Fig. 12 and dotted curves in Fig. 10 and Fig. 7) would be of any interest to the reader. They might be dropped, and some of the figures might be put together under one caption.

The manuscript would be publishable in the journal after such revision is properly made.

Referee: 2

Comments to the Author(s)

The authors have addressed the problem of entanglement Swapping for potential application of quantum networking for a distance of 5-10 km using a discrete variable (Fock state) and a continuous variable (cat state superposition) entangled state. The topic looks interesting and the results produced are well explained to understand the problem. The authors concluded that their suggested protocol is tolerant only to low levels of photonic losses. Acknowledging the contributions of the paper the reviewer has the following points which require attention or more explanation from the authors.

1. In the introduction section of the paper, more references are needed to highlight existing ES techniques along with their merits and demerits.
2. How the proposed Scheme overcome disadvantages of the previous schemes? (if there is any) Please clearly describe your contributions against those findings.
3. Its better to write conclusion in paragraph instead of bullets.
4. Is it possible to generate any level of entanglement using the proposed entanglement swapping protocol or it will only generate a state of some fixed degree of entanglement?

Board Member:

Comments to Author(s):

Thank you for your submission, which has now been reviewed. Both reviewers recommend updates to your manuscript, which one recommending a major update. All the suggestions seem good and would provide enhancements to your manuscript, hence please work through all the comments and I look forward to receiving your updated manuscript that contributes to the strategically important and growing area of quantum communications.

RSPA-2020-0237.R1 (Revision)

Review form: Referee 1

Is the manuscript an original and important contribution to its field?

Acceptable

Is the paper of sufficient general interest?

Acceptable

Is the overall quality of the paper suitable?

Good

Can the paper be shortened without overall detriment to the main message?

Yes

Do you have any ethical concerns with this paper?

No

Recommendation?

Accept as is

Comments to the Author(s)

The authors have addressed basically all the points in my previous report and revised/improved their manuscript accordingly. Although the authors' scheme is not immediately experimentally feasible, it adds some interesting knowledge to the field and suggests some research directions on the topic. I would recommend publication of this manuscript.

Review form: Referee 2

Is the manuscript an original and important contribution to its field?

Good

Is the paper of sufficient general interest?

Good

Is the overall quality of the paper suitable?

Good

Can the paper be shortened without overall detriment to the main message?

Yes

Do you think some of the material would be more appropriate as an electronic appendix?

No

Do you have any ethical concerns with this paper?

No

Recommendation?

Accept as is

Comments to the Author(s)

The reviewer is pleased to accept this article for publication in PRSA.

Decision letter (RSPA-2020-0237.R1)

13-Oct-2020

Dear Dr Parker

I am pleased to inform you that your manuscript entitled "Photonic Hybrid State Entanglement Swapping using Cat State Superpositions" has been accepted in its final form for publication in Proceedings A.

Our Production Office will be in contact with you in due course. You can expect to receive a proof of your article soon. Please contact the office to let us know if you are likely to be away from e-mail in the near future. If you do not notify us and comments are not received within 5 days of sending the proof, we may publish the paper as it stands.

Please note that as you have opted for open access then payment will be required before the article is published – payment instructions will follow shortly.

Under the terms of our licence to publish you may post the author generated postprint (ie. your accepted version not the final typeset version) of your manuscript at any time and this can be made freely available. Postprints can be deposited on a personal or institutional website, or a recognised server/repository. Please note however, that the reporting of postprints is subject to a media embargo, and that the status the manuscript should be made clear. Upon publication of the definitive version on the publisher's site, full details and a link should be added.

You can cite the article in advance of publication using its DOI. The DOI will take the form: 10.1098/rspa.XXXX.YYYY, where XXXX and YYYY are the last 8 digits of your manuscript number (eg. if your manuscript number is RSPA-2017-1234 the DOI would be 10.1098/rspa.2017.1234).

For tips on promoting your accepted paper see our blog post:
<https://royalsociety.org/blog/2020/07/promoting-your-latest-paper-and-tracking-your-results/>

On behalf of the Editor of Proceedings A, we look forward to your continued contributions to the Journal.

Sincerely,
Raminder Shergill
proceedingsa@royalsociety.org

on behalf of
Dr Ben Allen
Board Member
Proceedings A

Reviewer(s)' Comments to Author:

Referee: 2

Comments to the Author(s)

The reviewer is pleased to accept this article for publication in PRSA.

Referee: 1

Comments to the Author(s)

The authors have addressed basically all the points in my previous report and revised/improved their manuscript accordingly. Although the authors' scheme is not immediately experimentally feasible, it adds some interesting knowledge to the field and suggests some research directions on the topic. I would recommend publication of this manuscript.